# Construction of a Triple-Gene Deletion Mutant of Orf Virus and Evaluation of Its Safety, Immunogenicity and Protective Efficacy

**DOI:** 10.3390/vaccines11050909

**Published:** 2023-04-28

**Authors:** Zhanning Shen, Bo Liu, Zhen Zhu, Jige Du, Zhiyu Zhou, Chenfan Pan, Yong Chen, Chunsheng Yin, Yufeng Luo, Huanrong Li, Xiaoyun Chen

**Affiliations:** 1Animal Science and Techology College, Beijing University of Agriculture, Beijing 102208, China; 2China Institute of Veterinary Drug Control, Beijing 100081, China; 3International Atomic Energy Agency, Vienna International Centre, P.O. Box 100, A-1400 Vienna, Austria

**Keywords:** ORFV, ORFV 121, recombinant virus, triple-gene deletion, safety, protective efficacy, vaccine

## Abstract

Contagious ecthyma is a zoonotic disease caused by the orf virus (ORFV). Since there is no specific therapeutic drug available, vaccine immunization is the main tool to prevent and control the disease. Previously, we have reported the construction of a double-gene deletion mutant of ORFV (rGS14ΔCBPΔGIF) and evaluated it as a vaccine candidate. Building on this previous work, the current study reports the construction of a new vaccine candidate, generated by deleting a third gene (gene 121) to generate ORFV rGS14ΔCBPΔGIFΔ121. The in vitro growth characteristics, as well as the in vivo safety, immunogenicity, and protective efficacy, were evaluated. RESULTS: There was a minor difference in viral replication and proliferation between ORFV rGS14ΔCBPΔGIFΔ121 and the other two strains. ORFV rGS14ΔCBPΔGIFΔ121 induced continuous differentiation of PBMC to CD4^+^T cells, CD8^+^T cells and CD80^+^CD86^+^ cells and caused mainly Th1-like cell-mediated immunity. By comparing the triple-gene deletion mutant with the parental strain and the double-gene deletion mutant, we found that the safety of both the triple-gene deletion mutant and the double-gene deletion mutant could reach 100% in goats, while the safety of parental virus was only 50% after continually observing immunized animals for 14 days. A virulent field strain of ORFV from an ORF scab was used in the challenge experiment by inoculating the virus to the hairless area of the inner thigh of immunized animals. The result showed that the immune protection rate of triple-gene deletion mutant, double-gene mutant, and the parental virus was 100%, 66.7%, and 28.6%, respectively. In conclusion, the safety, immunogenicity, and immune-protectivity of the triple-gene deletion mutant were greatly improved to 100%, making it an excellent vaccine candidate.

## 1. Introduction

Contagious ecthyma, also known as orf, is a contact, epitheliophilic, and highly contagious zoonotic disease caused by parapoxvirus [1]. The disease is mainly prevalent in goats and sheep with clinical signs such as pustules, ulcers, scabs, and other lesions on the animal’s mouth and lips, which seriously affect the animal’s feeding situation [2]. Generally, the disease is a mild self-limiting disease. The lesions generally heal on their own within 1 to 2 months [3]. However, the long-term impact on feeding causes malnutrition in sheep and can result in secondary pathogenic infections, resulting in severe weakness and even the death of small ruminants. According to reports, the disease can be transmitted not only to humans [4,5], but also to deer, guinea pigs, dogs and other animals [6,7,8].

ORF is caused by the orf virus, a DNA virus belonging to the genus Parapoxvirus, family Poxviridae, with a total length of approximately 138 kbp that encodes approximately 132 genes [9,10]. The viral genome has a highly conserved region in the middle and variable regions at both ends, the same as the common pattern of the pox virus [11]. The middle region of the virus contains mainly essential genes encoding proteins involved in viral replication and viral structures, while the variable regions at both ends encode proteins related to host range, pathogenesis and virulence [12]. Among those genes, the B2L gene of ORFV is commonly adopted as a target for polymerase chain reaction (PCR) detection of ORFV for phylogenetic analysis due to its specificity and degree of conservation [13]. The polypeptide encoded by the highly conservative gene, F1L, is frequently used in serological diagnosis of orf in sheep and goats [14]. Notably, ORFV encodes many immune-evasion regulatory proteins (IERPs) that modulate the pro-inflammatory response to host infection. Among the IMPs described so far, chemokine-binding protein (CBP; ORFV112) [15], the soluble protein inhibitors sheep granulocyte monocyte colony-stimulating factor (GM-CSF) and interleukin 2(IL-2) (GIF, ORFV117) [2,12], and the sheep interleukin-10 (vIL-10; ORFV127) [16] were characterized based on ORFV strain NZ2, an extensively studied ORFV strain from New Zealand [17]. One of the main functions of CBP is to facilitate immune evasion by high-affinity binding and inhibiting the production of cytokines and thus inhibiting the capture of viruses by immune cells [18]. GIF has dual activities of inhibiting host GM-CSF and IL-2 [18]. The gene encoding the ORFV 121 protein is located in the 3′ reverse terminal repeat sequence region of the ORFV genome [19]. ORFV 121 can successfully evade the host immune system by directly binding to NF-κB-p65 and inhibiting its phosphorylation and nuclear translocation processes, thereby inhibiting the transcription and translation of immune-related genes and weakening the host immune response [20]. In addition, the deletion of ORF 121 can result in a significant reduction in ORFV virulence [19].

ORFV has been proposed as a vaccine vector due to its limited host range, lack of virus-neutralizing antibodies, large genome, and broad immunomodulatory properties [21,22]. The conventional approach to controlling ORFV is the prophylactic application of vaccination. Among commercially available vaccines, inactivated vaccines do not prevent the host from being infected, while live attenuated vaccines and recombinant live vaccines provide effective protection against ORFV [23]. Live attenuated ORFV vaccines have been the mainstay of prevention of contagious ecthyma in sheep and goats for decades, but they cannot consistently induce durable immunity [24]. In addition, the risk of reversion to virulence has been reported for successive generations of live vaccines. Recombinant live attenuated vaccines generally use genetic engineering techniques to knock out virulence genes of viruses, to achieve a reduction in the virulence of the pathogen itself and improve its safety without changing the immunogenicity of the virus itself. Recent studies have shown that deletion of some virulence genes of ORFV can reduce its virulence and produce live attenuated vaccines with good immunoprotective properties [25,26]. Compared to traditional live attenuated vaccines, live attenuated vaccines prepared by genetic engineering with deletion of virulence genes have a higher safety profile and do not suffer from virulence reversion. Currently, the use of genetic engineering to prepare live attenuated ORFV vaccines is the main research direction.

Based on the successful construction of the double-gene deletion mutantrGS14ΔCBPΔGIF [3] in our previous study, a triple-gene deletion mutant rGS14ΔCBPΔGIFΔ121 was constructed by deleting gene 121. The new mutant’s in vitro growth and replication, immunogenicity, and immuno-protection rate were investigated to evaluate its safety and efficacy. The result indicated that the triple-gene deletion mutant has better immune protectivity with better safety and immunogenicity. The new mutant could trigger the continuous high level of differentiation of PBMC to CD4^+^ T cells, CD8^+^ T cells and CD80^+^CD86^+^ cells and mainly cause Th1-like immune responses.

## 2. Materials and Methods

### 2.1. Cells and Viruses

ORFV-GS14 preserved in our laboratory (an ORFV isolated from Gansu Province, China, designated as GS14) was used as a parent virus throughout the experiment and as a genetic background virus to generate target gene-deleted virus. Virus stocks were propagated in Primary Goat Testicular cells (PGT cells) and stored at −80 °C. The wild-type virus was prepared by sampling scabs from the goat and grinding them after the challenge experiment [3]. The cbp/gif double-gene deletion mutant (rGS14ΔCBPΔGIF) of ORFV was constructed in our laboratory [3]. The EGFP-N1 and other plasmids were kindly donated by the China Institute of Veterinary Drug Control. The PGT cells were prepared and frozen in our laboratory [3,27]. Sheep testis cells were cultured in Ham’s DMEM/12 Gluta MAX medium (Thermo Fisher Scientific, USA) containing 10% fetal bovine serum (FBS) (Every Green, Zhejiang Tianhang Biotechnology Co., PGTd, Hangzhou, China) and 1% penicillin-streptomycin solution (with a final concentration of 100 IU/mL of penicillin and 100 µg/mL of streptomycin) at 37 °C with 5% CO_2_.

### 2.2. Construction of rGS14ΔCBPΔGIFΔ121

We mainly used the principle of homologous recombination for the deletion of the 121 gene (GenBank: OM687301.1) of rGS14ΔCBPΔGIF (Figure 1). In the process of constructing the deletion vector, the left homologous arm of 121 gene, the eGFP reporter gene and the right homologous arm were mainly ligated using fusion PCR, and transfection was performed using Shanghai Biotech after sequencing. PGT cells were spread evenly in 6-well plates, treated with 0.5 mL 10^7.7^ TCID50/mL of rGS14ΔCBPΔGIF, and subsequently transfected with 4 mg of the transfer vector p121hm1-GFP-121hm2 at 24 h post-infection (hpi) using 5 mL of Lipofectamine 3000 reagent (Thermo Fisher Scientific, Waltham, MA, USA). The green fluorescent areas were picked and purified using the limited dilution method. Further purification was performed using flow cytometry until all the lesioned areas within the field of view fluoresced. Afterward, the entire DNA of the purified virus was extracted with the AxyPrep body fluid viral DNA/RNA miniprep kit (Axygen, Union City, CA, USA). The 121 gene was identified using PCR amplification, and no band appeared as a successful deletion, and rGS14ΔCBPΔGIFΔ121-eGFP was obtained.

After that, the left homologous arm and the right homologous arm of the 121 gene were ligated using fusion PCR to construct the vector for transfection, and after sequencing without errors, they were transfected into LT cells in 6-well plates with access to rGS14ΔCBPΔGIFΔ121-eGFP length, and the areas with lesions that did not fluoresce were picked, repeatedly frozen and thawed, and then purified using the limited dilution method and flow cytometry. After purification to complete non-fluorescence in green, deletion identification was performed with primers and sent for sequencing to obtain rGS14ΔCBPΔGIFΔ121.

**Figure 1 vaccines-11-00909-f001:**
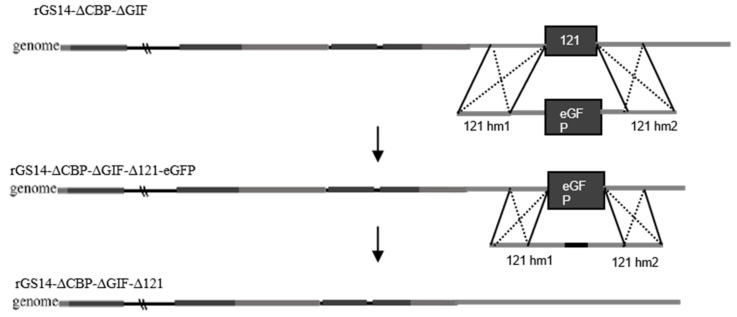
Schematic diagram of rGS14ΔCBPΔGIFΔ121 construction.

### 2.3. Animals

The animals used in this study were 1-month-old ORFV-negative goats, and a total of 47 goats were used, all purchased from Honest Farm, Mancheng District, Hebei, China. The goats were randomly grouped. The animal procedures were approved by the Animal Care and Use Committee of China Institute of Veterinary Drug Control.

### 2.4. Virus Titration and Growth Curve with TCID50 Titration

Following the conventional TCID_50_ assay, rGS14ΔCBPΔGIFΔ121, rGS14ΔCBPΔGIF and GS14 were diluted by a gradient from 10^−1^ to 10^−10^ and added into a 96-well plate seeded with PGT cells. The viruses were removed after two hours of adsorption at 37 °C and 5% CO_2_. After that, PBS was added and used to wash the cells and repeated twice. The medium was replaced with 4% fetal bovine serum maintenance medium, and cells were incubated at 37 °C with 5% CO_2_. Calculations were performed according to the Reed-Muench method after 7 days of incubation post-infection.

In order to investigate the cytopathic effect (CPE) differences among GS14-DCBP-DGIF-D121, rGS14ΔCBPΔGIF and GS14, we used the same TCID_50_ for inoculation and observed and recorded the CPE characteristics at different time points of dispersion.

Equal TCID_50_ doses of rGS14ΔCBPΔGIFΔ121, rGS14ΔCBPΔGIF and GS14 viruses were inoculated into 6-well plates with full confluency of PGT cells. Cells were incubated at 37 °C with 5% CO_2_. The viral fluid was collected at 2 h, 4 h, 8 h, 12 h, 24 h, 36 h, 48 h, 60 h, and 72 h post-infection. The viral fluid was frozen and thawed three times. Then, the lysate was titrated for viral load at each time point by TCID_50_ and quantitative PCR (qPCR).

To investigate the genetic stability of rGS14ΔCBPΔGIFΔ121, we performed consecutive virus passaging in PGT cells. PGT cells were infected with low MOI (~1) recombinant virus and harvested 7 h post infection. The virus stability of the 121 gene deletion on the 1st, 5th, 10th and 15th passage was evaluated by PCR.

### 2.5. Safety Test

To assess the virulence of the gene-deficient recombinant virus on goats, 16 antibody-negative one-month-old goats were divided into four groups and inoculated with rGS14ΔCBPΔGIFΔ121 (s goats per group) (hereafter referred to as rGS14ΔCBPΔGIFΔ121 group), GS14 (3 goats per group) (hereafter referred to as GS14 group), PBS (3 goats per group) (hereafter referred to as PBS group)and rGS14ΔCBPΔGIF (3 goats per group) (hereafter referred to as rGS14ΔCBPΔGIF group) at 0.2 mL 10^6^TCID_50_ each via inner thigh scratch inoculation. The animals were photographed on days 5, 7 and 14 post-inoculation. Clinical signs were scored by 6 persons unrelated to this study every 2 days from the inoculation day. Clinical scores were performed according to Martins’ method [28]. Four clinical indicators, namely hyperemia, vesicles and/or pustules, scabs and exudation and/or bleeding, were used as criteria and scored according to severity (see supplementary Appendix A for rating scale). The higher the score, the more severe the clinical signs were. The clinical scores were calculated for each animal, and then the mean clinical score and standard deviation of the mean (mean ± standard error of the mean (SEM)) were calculated for each group.

### 2.6. Flow Cytometry Analysis

Blood was collected on days 3, 5, 7 and 14 post-inoculation to determine CD4^+^, CD8^+^ T cells and CD80^+^CD86^+^ dendritic cell differentiation using flow cytometry. The anticoagulant containing EDTA was added to the whole blood and mixed. A 100 μL mixture was taken and added with the corresponding antibodies (CD4 Monoclonal Antibody (44.38), PE, Thermo; CD8 Monoclonal Antibody (38.65), FITC, Thermo; CD86 antibody|IL-A190, Bio-rad; CD80 antibody|IL-A159, Bio-rad, Hercules, CA, USA) following recommended antibody instructions. The sample was vortexed and then incubated for 15 min without light. Then, the sample was added to 2 mL of hemolysin containing an immobilizer. After vortexing and incubation for 10 min, the sample was centrifuged at 1500 rpm for 5 min to discard the supernatant. Samples were washed again with PBS and finally resuspended with 200 μL PBS and analyzed using flow cytometry (BD FACSAria II).

### 2.7. Cytokines Detection and Analysis

The expression of cytokines, including IL-2, IL-4, IL-10 and IFN-γ, was also examined on day 3, 5, 7 and 14 post-inoculation using commercial kits (Goat Interleukin 10, IL-10 ELISA Kit, Goat Interleukin 2(IL-2) ELISA Kit, Goat Interleukin 4, IL-4 ELISA Kit, Goat Interferon γ, IFN-γ ELISA Kit, Cusabio Biotech Co., PGTD, Wuhan, China).

### 2.8. Comparison of Immune Protection of rGS14ΔCBPΔGIFΔ121

The goats were grouped according to rGS14ΔCBPΔGIFΔ121 (11 goats per group) (hereafter referred to as G1), rGS14-ΔCBP-ΔGIF (3 goats per group) (hereafter referred to as G2), GS14 (7 goats per group) (hereafter referred to as G3), and PBS group (10 goats per group) (hereafter referred to as G4) and inoculated with a scratch on the left side of the thigh in the hairless area, and the inoculation dose is 0.2 mL 10^6^TCID_50_. After 14 days, the goats’ homology was challenged on the right side of the thigh in the inner hairless area with the wild-type strain ORFV isolated from the scabs of the infected goat. The strain was preserved in our laboratory and the main virulence genes, such as vIL-10, vir, cbp, gif and 121, were sequenced by Zhu et al. in 2022 [3]. Sequencing analysis based on the B2L gene showed that the nucleotide identity of wild-type ORFV with ORFV-GS14 strain was 97.09% [3]. The minimum pathogenic dose of this virus was experimentally demonstrated to be 10^−2^ dilution. The virus was challenged according to the minimum pathogenic dose, observed, photographed and recorded for 14 days, and then scored according to the scoring criteria. Scabs from the challenging site and ORFV-susceptible tissues from the muzzle area were taken at the end of the observation period. Ten percent suspensions of tissue samples were ground, after which the viral load of ORFV was determined using the fluorescent quantitative PCR method previously established in our laboratory.

### 2.9. Statistical Analysis

Statistical analysis of cytokine and PBMC differentiation data was performed using two-way analysis of variance by GraphPad Prism version 8.0 (GraphPad Software Inc., San Diego, CA, USA). Statistically significant was determined when *p* < 0.05 or *p* < 0.01.

## 3. Result

### 3.1. Construction of Recombinant Virus rGS14ΔCBPΔGIFΔ121

The 121 gene and the sequence of the left and right homologous arms of recombinant rGS14ΔCBPΔGIF were determined. The 121 gene was successfully replaced with the GFP gene. After four rounds of purification, all the cytopathic areas of the virus emitted green fluorescence (Figure 2A), which indicated all of the 121 genes were successfully replaced with the GFP gene. After further continuous purification, no green fluorescence was observed in the lesioned areas under fluorescence microscopy (Figure 2B). Sequencing results showed no mutation in the rest of the virus, except for the missing part of the gene (see supplementary Appendix A for partial results graphs).

### 3.2. In Vitro Characterization of the Recombinant Virus

The proliferation characteristics of recombinant strains rGS14ΔCBPΔGIFΔ121 and rGS14ΔCBPΔGIF and parental strain GS14 were compared in PGT cells. The rGS14ΔCBPΔGIFΔ121 was 10^−7^/0.1 mL, which was similar to that of the parental ORFV-GS14 (10^−6.7^/0.1 mL) and rGS14ΔCBPΔGIF (10^−6.2^/0.1 mL). There was no statistical difference in the CPE produced at different time points after infection among the three viruses (Figure 3A). The in vitro growth kinetics of rGS14ΔCBPΔGIFΔ121, rGS14ΔCBPΔGIF and GS14 in PGT cells were measured and compared at a dose of MOI=1. Results showed similar growth kinetics (Figure 3B). Afterward, the viral genomic DNA of the three viruses was extracted and quantified by qPCR, which also indicated similar viral growth trends (Figure 3C).

### 3.3. rGS14ΔCBPΔGIFΔ121 Group < rGS14ΔCBPΔGIF Group < GS14 Group in Terms of Virulence

After immunization, the rGS14ΔCBPΔGIFΔ121 group, the rGS14ΔCBPΔGIF group and the PBS group showed bleeding, redness and swelling in the first two days, and the wounds gradually healed with time. There was minimal inflammatory reaction due to the feeding environment, which was healed afterwards. The GS14 group started to show pustules and herpes on day 7 in addition to bleeding, redness and swelling in the first period and then started to exhibit scabs, which were only partially healed by the 14th day (Figure 4A). By scoring the immunization sites for assessment, the rGS14ΔCBPΔGIFΔ121 group, rGS14ΔCBPΔGIF group and PBS group had lower scores on day 14 after immunization, while the GS14 group had the highest score. (Figure 4B).

### 3.4. rGS14ΔCBPΔGIFΔ121 Causing High Levels of Expression of CD4^+^ Cells, CD8^+^ Cells and CD80^+^CD86^+^ Cells

In addition to the clinical signs, we also analyzed the immune response induced by rGS14ΔCBPΔGIFΔ121. According to the literature, ORFV mainly causes cellular immune responses [12], so we chose to detect cell subpopulations such as CD4^+^ cells, CD8^+^ cells and CD80^+^CD86^+^ cells (Figure 5). Flow cytometry analysis of whole blood from the immunized goat showed that rGS14ΔCBPΔGIFΔ121 rapidly caused differentiation of PBMC to CD4^+^ cells, CD8^+^ cells and CD80^+^CD86^+^ cells. In terms of CD4^+^ cell subpopulations, rGS14ΔCBPΔGIFΔ121 caused an increase in the number of CD4^+^ cells on day 3 and maintained a consistently high level of expression (*p* < 0.0001), while that of rGS14ΔCBPΔGIF started to rise on day 5 and maintained a high level of expression after reaching a peak on day 7 (*p* < 0.0001). The number of the CD4^+^cell subpopulation of the rGS14ΔCBPΔGIFΔ121 group was consistently higher than that of the rGS14ΔCBPΔGIF group in the first 7 days after immunization (*p* < 0.05) (Figure 5B). In terms of CD8+cell subpopulation, the rGS14ΔCBPΔGIFΔ121 group induced an increase on day 5 and then maintained at a high level (*p* < 005), while that of the rGS14ΔCBPΔGIF group peaked on day 7 (*p* < 0.05) and then decreased (Figure 5C). rGS14ΔCBPΔGIFΔ121, rGS14ΔCBPΔGIF and GS14 all caused differentiation of PBMC to CD80^+^CD86^+^ cells on day 3. rGS14ΔCBPΔGIFΔ121 group peaked on day 5 (*p* < 0.0001) and then started to decline. The rGS14ΔCBPΔGIF group peaked on day 7 (*p* < 0.0001) and then declined. The GS14 group peaked on day 5 (*p* < 0.0001) and then declined (Figure 5E).

### 3.5. rGS14ΔCBPΔGIFΔ121 Induces High Levels of IL-2, IL-10 and IFN-γ Expression

To further investigate the immune response induced by rGS14ΔCBPΔGIFΔ121, we examined the expression of IL-2, IL-10, IL-4 and IFN-γ (Figure 6). Regarding IL-2, rGS14ΔCBPΔGIFΔ121 was consistently expressed at a high level during the first 7 days (*p* < 0.001), starting to rise on day 3 (*p* < 0.0001) and reaching a peak on day 5 (*p* < 0.0001). The expression level of IL-2 in rGS14ΔCBPΔGIFΔ121 group was always higher than that in the GS14 group during the first 7 days (*p* < 0.05). GS14 caused a high level of IL-2 expression on day 14 (*p* < 0.05) (Figure 6A). Detection of IL-10 expression levels revealed that animals in the rGS14ΔCBPΔGIFΔ121 group started to increase in IL-10 expression levels after immunization (*p* < 0.01), reached a peak on day 5 (*p* < 0.0001) and gradually decreased after day 7 (*p* < 0.001). The rGS14ΔCBPΔGIFΔ121 group consistently expressed high levels of IL-10 during the first 7 days (*p* < 0.001) and then decreased (Figure 6B). In terms of IFN-γ, rGS14ΔCBPΔGIFΔ121 could cause high levels of expression in goats on days 5 and 7 (*p* < 0.0001) (Figure 6C). By detecting IL-4, rGS14ΔCBPΔGIFΔ121 caused high-level expression in goats only at day 7 (Figure 6D). Taken together, it was tentatively determined that rGS14ΔCBPΔGIFΔ121 mainly caused Th1-like cellular immune responses.

### 3.6. rGS14-TrypMut > rGS14-DoubMut > GS14-w t > PBS in Terms of Protective Efficacy

Protective efficacy is an important parameter of a vaccine. We investigated the immune-protectivity properties of different live vaccines by using wild-type strain obtained from scabs of infected goats after immunizing the animals. Within 14 days after challenging, all members of the control group developed pustules, papules and scabs, which proved the validity of this experiment. In contrast, the goats in G1 showed bleeding, redness and swelling on day 2 post-challenge. Clinical signs were gradually alleviated over time and almost completely subsided by day 7. Only a few goats still had mild inflammatory reactions, which disappeared by day 14. The immuno-protection rate could reach 100%. G2 and G3 groups showed pustules, papules and other signs on day 7 after the immunization. Some goats still showed scabs and other signs by day 14, so the protection rate could not reach 100% (Figure 7A). The clinical scoring results showed PBS>GS14-wt=rGS12-DoubMut>rGS-TrypMut (Figure 7B). The qPCR results showed that G1 had the lowest ORFV viral load regardless of the challenging site or the susceptible site of the mouth and lips, with the mean lgcopies of the challenging site at 4 and the mean lgcopies of the mouth and lips site at 2.87. In the G2 group, the mean lgcopies value of ORFV at the challenging site was 5.97, and the mean lgcopies value at the mouth and lip site was 4.94. The control group had the highest ORFV load: the mean lgcopies value of ORFV at the challenging site was 6.82 and the mean lgcopies value at the mouth and lip site was 6.26. By comparison, the overall ORFV load of the challenged animals was PBS>rGS14-DoubMut>rGS14-TrypMut. (Figure 7C).

## 4. Discussion

ORF is prevalent worldwide as a zoonotic disease, and it has caused huge economic losses to the sheep industry [29]. However, for ORFV, the main commercially available vaccines are live scab-based vaccines and live attenuated cell-culture-based vaccines [30]. Although the above-mentioned two vaccines play an important role in the fight against ORFV [31], live scab-based vaccines are less safe [32]. Live cell-culture-based vaccines are relatively more popular, but can provide limited protection [33]. Recent research updates have shown that recombinant live attenuated vaccines have promising applications. Genetically attenuated strains obtained by knocking out the virulence gene of the vaccine strains provide better safety and significantly improve immunogenicity compared to conventional vaccines [27,32,34]. Due to the large genome of ORFV and the numerous virulence genes, obtaining attenuated vaccine strains by knocking out their virulence genes has become the main method of vaccine preparation [35].

In this study, we constructed a triple-gene-deletion recombinant strain rGS14ΔCBPΔGIFΔ121 based on the double-gene-deletion recombinant strain rGS14ΔCBPΔGIF constructed in the previous study [3] and explored the effects and mechanisms of deletion of CBP, GIF and the 121 genes in the pathogenesis of ORFV. It has been shown that the 121 gene mainly binds to NF-κB-p65 in the cytoplasm, thereby inhibiting its phosphorylation and translocation to weaken the host immune response [19]. The 121 gene encodes the virulence protein of ORFV, the knockdown of which does not affect the proliferation and replication of the virus itself [19]. In the present study, we compared the in vitro growth characteristics of ORFV-GS14, rGS14ΔCBPΔGIF and rGS14ΔCBPΔGIFΔ121. The comparison of their cytopathic morphology and TCID50 growth curves showed no significant difference among the three strains, which was consistent with the previous result for rGS14ΔCBPΔGIF and ORFV-GS14 [3]. This also indicated that all three genes, CBP, GIF and 121, were not associated with viral replication and proliferation, which was consistent with the previous findings. In addition, we found that rGS14ΔCBPΔGIFΔ121 had good genetic stability after consecutive passaging, which was conducive to the large-scale preparation of recombinant viral strains.

For vaccines, safety and stability are the two important indicators. In this study, we minimized the risk of virulence reversion and mutation by knocking out the virulence gene. The safety of GS14, rGS14ΔCBPΔGIF and rGS14ΔCBPΔGIFΔ121 was evaluated by vaccinating 1-month-old goats. The results showed that both rGS14ΔCBPΔGIF and rGS14ΔCBPΔGIFΔ121 met the safety criteria. In the differentiation assay of PBMC, all three viruses could induce high expression of CD80^+^CD86^+^ double-positive cells, but rGS14ΔCBPΔGIF and rGS14ΔCBPΔGIFΔ121 induced higher and more expression with a longer-lasting effect. This might be related to the deletion of CBP and GIF, which could weaken the immune evasion of the virus [26,36]. By comparative analysis, we saw that the rGS14ΔCBPΔGIFΔ121 group peaked much earlier, at day 5, and far exceeded the rGS14ΔCBPΔGIFΔ121 group and GS14 group and remained at high levels of expression, which might cause ORFV to be recognized and captured by DC cells earlier and elicit an earlier immune response. The level of CD80^+^CD86^+^ cells induced by rGS14ΔCBPΔGIF consistently showed a slow increase, suggesting that rGS14ΔCBPΔGIF also greatly weakened immune evasion. Among the variations in the horizontal expression of CD4^+^ T cells and CD8^+^ T cells, both rGS14ΔCBPΔGIFΔ121 and rGS14ΔCBPΔGIF induced higher levels of expression, which might be related to the intrinsic properties of ORFV. The immune responses elicited by both rGS14 and rGS14ΔCBPΔGIFΔ121 appeared earlier after weakening the immune evasion. GS14 group did not show the immune responses elicited by ORFV during the 14-day experiment due to its immune evasion and individual differences between animals. In the CD4^+^ T cell assay, the overall expression level of rGS14ΔCBPΔGIFΔ121 was consistently higher than that of rGS14ΔCBPΔGIF during the first 7 days and was significantly different from that of the rGS14ΔCBPΔGIF group, indicating that rGS14ΔCBPΔGIFΔ121 caused a much higher level of cellular immune. In the CD8^+^ T cell assay, rGS14ΔCBPΔGIF caused more significant changes than rGS14ΔCBPΔGIF in goats at day 5. rGS14ΔCBPΔGIFΔ121 consistently maintained high levels of CD8^+^cell expression within 14 days. We suggest that the deletion of the 121 gene greatly diminishes the virulence and immune evasion of ORFV itself and that it can induce an earlier, stronger and longer-lasting immune response, which will greatly shorten the time for the host to react with an immune response when re-infected.

By further exploring the immune response induced by the recombinant virus rGS14ΔCBPΔGIFΔ121, we found that it could cause high levels of IL-2, IL-10 and IFN-γ expression. Among these, the high level of IL-10 expression was related to the characteristics of ORFV itself. As the virus proliferated, host cells continued to secrete IL-10, thereby suppressing the cellular immune response [37]. This suggests that the deletion of CBP, GIF and 121 does not affect the proliferation and replication of the virus itself. In addition, for the host this helps the host to provide a faster immune re-response in the face of a wild virus challenge. In the IL-4 expression level assay, it was observed that rGS14ΔCBPΔGIFΔ121 was only expressed at a high level on day 7, which might be due to individual differences in the goats themselves. It is noteworthy that either sustained activation or sustained inhibition of the NF-κB signaling pathway in the epidermis mediates a severe skin inflammatory response [38], which leads to increased secretion of IFN-γ and IL-2, with consistent experimental results. Analysis of the experimental results showed that rGS14ΔCBPΔGIFΔ121 could cause high levels of IL-2 and IFN-γ expression, which led to Th1-like cellular immune responses, as expected. These results suggest that the virulence and immune evasion of the gene-deficient recombinant viruses GS14, rGS14ΔCBPΔGIF, and rGS14ΔCBPΔGIFΔ121 gradually decayed, and rGS14ΔCBPΔGIFΔ121 was able to elicit a more intense immune response earlier than rGS14ΔCBPΔGIF.

In the assay to detect immune protection, we used wild-type ORFV on goats inoculated with recombinant virus GS14, rGS14ΔCBPΔGIF and rGS14ΔCBPΔGIFΔ121 separately. The results showed that the immune protection rates of triple-gene deletion mutant, double-gene mutant, and the parental virus were 100% (11/11), 66.7% (2/3), and 28.6% (2/7), respectively, which were different from the previous result [3]. This may be due to the individual differences in goats and the change in inoculation site. By testing the viral load of ORFV at susceptible sites in the challenged group, it was seen that the rGS14ΔCBPΔGIFΔ121 immunized group had the lowest viral load. The result indicates that the immune protective properties of the triple-gene deletion strain rGS14ΔCBPΔGIFΔ121 were superior to those of GS14 and rGS14ΔCBPΔGIF. This might be related to the deletion of the 121 gene. The 121 gene inhibits its transactivation activity by binding to NF-κB-p65 in the cytoplasm and inhibiting its phosphorylation and nuclear translocation [19]. NF-κ B signaling pathway mediates several aspects of skin homeostasis, including the innate immune response of keratinocytes [38,39]. The deletion of CBP and GIF weakened the immune evasion ability of the virus [3], making the virus easier to detect and capture, while the deletion of 121 made the host immune response more intense, allowing the host to eliminate the virus through the inflammatory response, strengthening the self-protection of the immune system.

## 5. Conclusions

In summary, the triple-gene deletion strain rGS14ΔCBPΔGIFΔ121 was successfully constructed in this study. The safety, immunogenicity and protective rates were evaluated. The result showed that the triple-gene deletion mutant strain rGS14ΔCBPΔGIFΔ121 could induce Th-1-like immune response and fully protect goats against the wild-type ORFV challenge. The prototype vaccine candidate, elicited earlier, increased and sustained immune responses.

## Figures and Tables

**Figure 2 vaccines-11-00909-f002:**
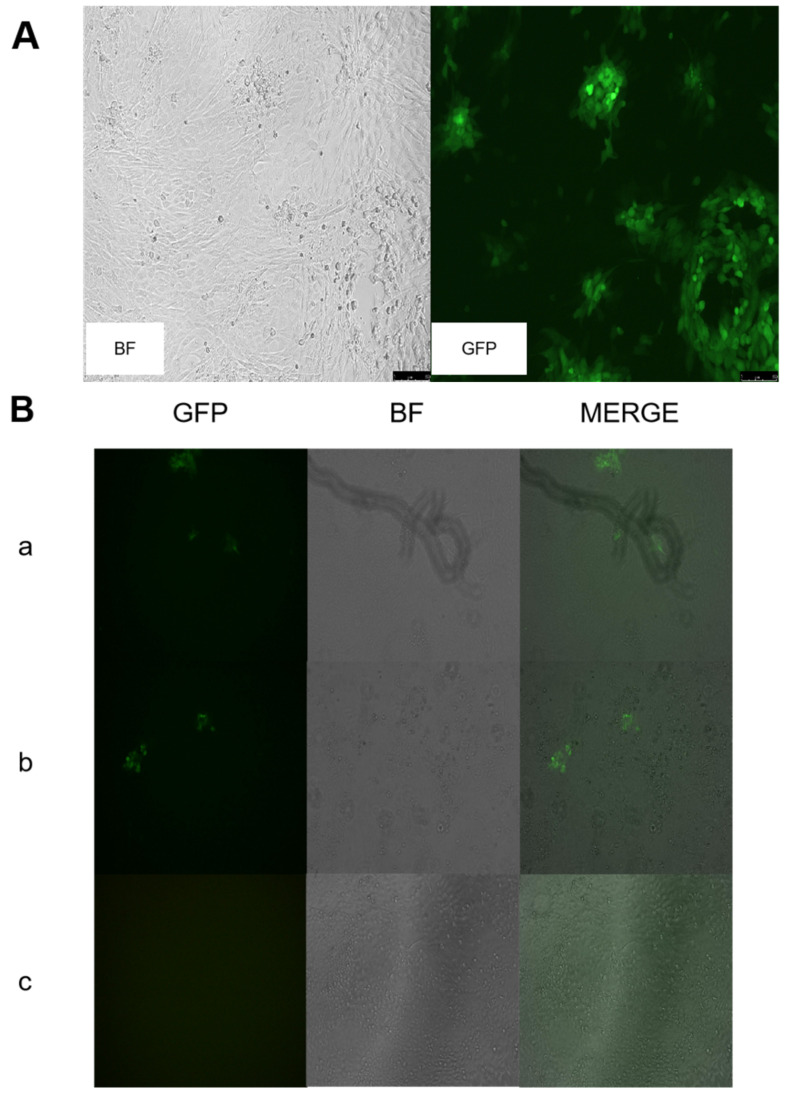
Construction of recombinant virus rGS14ΔCBPΔGIFΔ121: (**A**) rGS14ΔCBPΔGIFΔ121-GFP was fully purified. (**B**) Plague purification of rGS14ΔCBPΔGIFΔ121 (**a**): round 1; (**b**): round 3; (**c**): round 4.

**Figure 3 vaccines-11-00909-f003:**
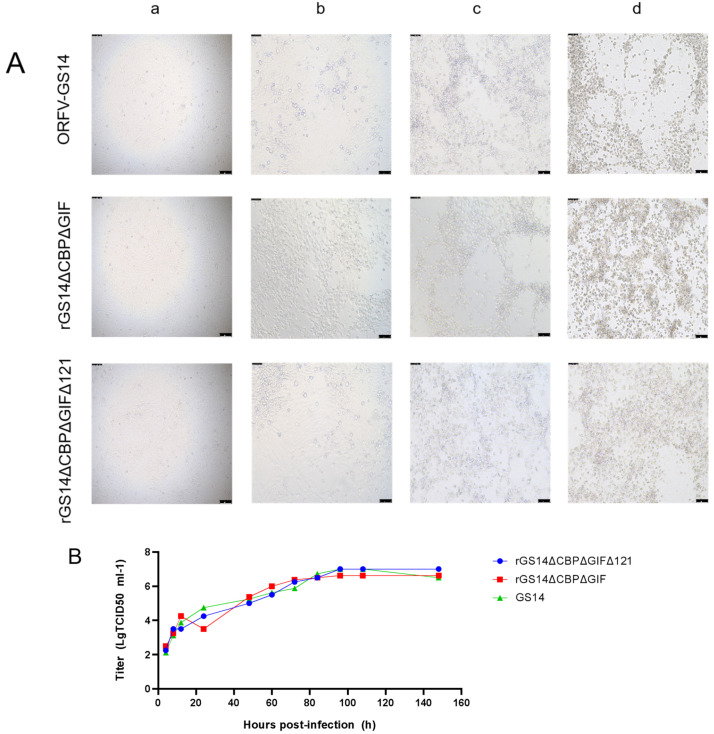
Comparison of in vitro growth curves and in vitro lesion morphology of parental strain ORFV-GS14, rGS14ΔCBPΔGIF and rGS14ΔCBPΔGIFΔ121: (**A**) In vitro CPE at different time points after infection with ORFV-GS14, rGS14ΔCBPΔGIF and rGS14ΔCBPΔGIFΔ121. The magnification of the microscope is 100×; (**a**) 24 h post-infection (**b**): 48 h post-infection (**c**): 72 h post-infection (**d**): 96h post-infection. (**B**) In vitro CPE of rGS14ΔCBPΔGIFΔ121, rGS14ΔCBPΔGIF and GS14 single-step growth curves in PGT cells (MOI = 1). (**C**) The number of viral infections at different time points after infection of rGS14ΔCBPΔGIFΔ121, rGS14ΔCBPΔGIF and GS14 in PGT cells. Calculations were based on qPCR detection of viral load at different time points.

**Figure 4 vaccines-11-00909-f004:**
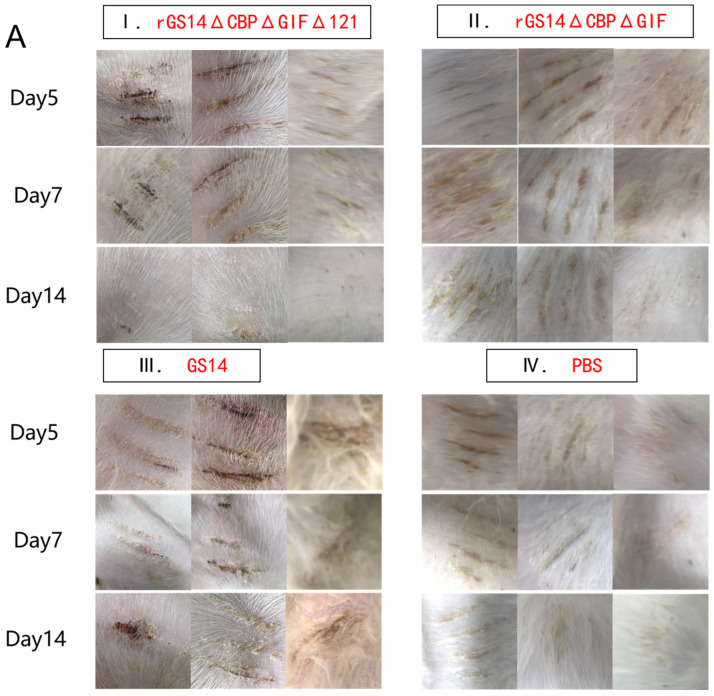
Comparison of the safety of rGS14ΔCBPΔGIFΔ121, rGS14ΔCBPΔGIF and GS14: (**A**) Clinical signs on days 5, 7 and 14 after immunization with rGS14ΔCBPΔGIFΔ121, rGS14ΔCBPΔGIF, GS14 and PBS, respectively. (**I**) clinical signs on days 5, 7 and 14 in 3 goats in the group immunized with rGS14ΔCBPΔGIFΔ121; (**II**) clinical signs on days 5, 7 and 14 in 3 goats in the group immunized with rGS14ΔCBPΔGIF; (**III**) clinical signs on days 5, 7 and 14 in 3 goats in the group immunized with GS14; (**IV**): clinical signs on days 5, 7 and 14 in 3 goats in the control group immunized with PBS; (**B**). A statistical line graph of the clinical signs scores of goats within 14 days after immunization with rGS14ΔCBPΔGIFΔ121, rGS14ΔCBPΔGIF, GS14 and PBS, with each data set expressed as mean standard error. Data for each group are expressed as mean standard error, * *p* < 0.05, ** *p* < 0.01, *** *p* < 0.001, **** *p* < 0.0001.

**Figure 5 vaccines-11-00909-f005:**
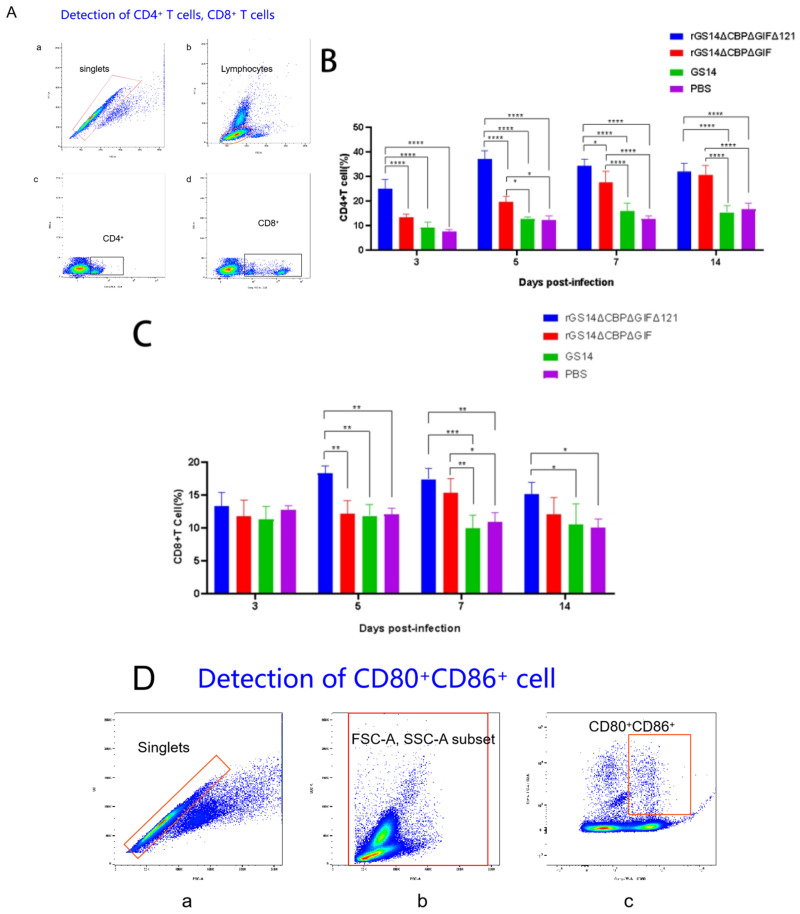
Statistical plots of cell subpopulation changes in the goat groups after immunization. Data for each group are expressed as mean standard error, * *p* < 0.05, ** *p* < 0.01, *** *p* < 0.001, **** *p* < 0.0001: (**A**) Schematic diagram of the process of sorting CD4^+^ T cells and CD8^+^ T cells in goats after immunization by flow cytometry: (**a**) removal of adhesions by FSC-A:FSC-H to obtain singlets; (**b**) gating of lymphocytes with small FSC and small SSC; (**c**) gating of CD4^+^ T cells; (**d**) gating of CD8^+^ T cells. (**B**) Whole blood was collected from the goats at different time points after immunization, and the percentage of CD4^+^ T cells in vivo was detected. (**C**) Whole blood was collected from the immunized goats at different time points, and the percentage of CD8^+^ T cells in the body was detected. (**D**) Schematic diagram of the process of CD80^+^CD86^+^ cells in the goats after immunization by flow cytometry: (**a**) removal of adhesions by FSC-A: FSC-H to obtain singlets; (**b**) all leukocytes were gated with FSC-A and SSC-A; (**c**): CD80+CD86^+^ cells were gated with scatter plots of CD80-PE and CD86-FITC. (**E**) Whole blood was collected from the goats at different time points after immunization, and the percentage of CD80^+^CD86^+^ cells in the body was detected.

**Figure 6 vaccines-11-00909-f006:**
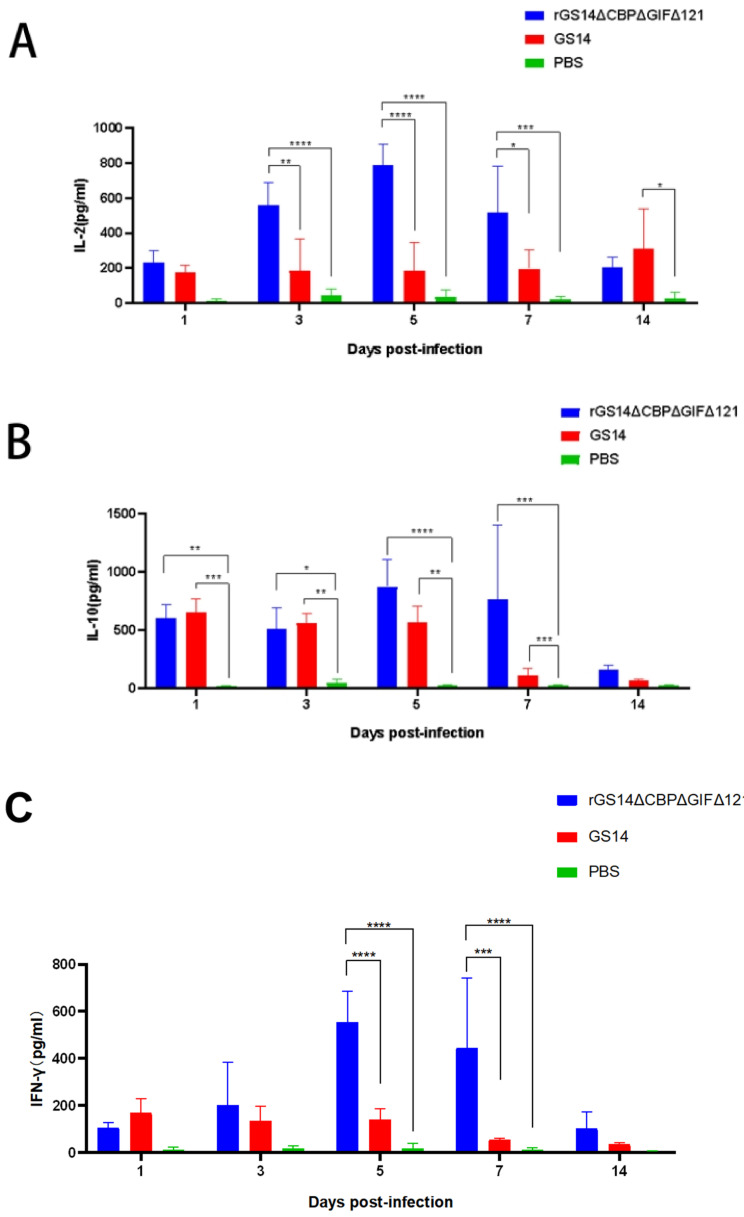
Changes in cytokine levels after inoculation with the rGS14ΔCBPΔGIFΔ121 group, GS14 group and PBS group. Data for each group are expressed as mean standard error, * *p* < 0.05, ** *p* < 0.01, *** *p* < 0.001, **** *p* < 0.0001. (**A**) Sera from goats of the different treatment groups were taken 14 days post inoculation and the levels of IL-2 determined. (**B**) Sera from goats of the different treatment groups were taken 14 days post-inoculation and the levels of IL-10 were determined. (**C**) Sera from goats of the different treatment groups were taken 14 days post inoculation, and the levels of IFN-γ determined. (**D**) Sera from goats of the different treatment groups were taken 14 days post-inoculation and the levels of IL-4 were determined.

**Figure 7 vaccines-11-00909-f007:**
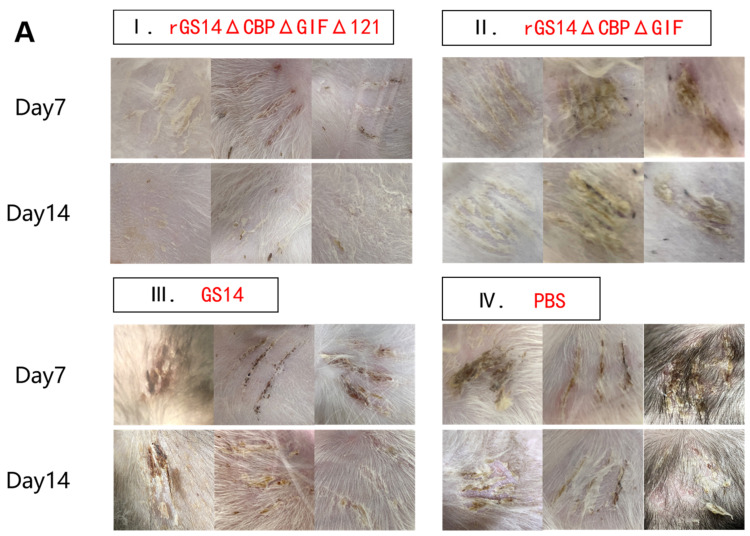
Clinical protective efficacy analysis and comparison in rGS14ΔCBPΔGIFΔ121, rGS14ΔCBPΔGIF, GS14 and PBS groups. (**A**) Plots of changes in clinical signs of goats in the rGS14ΔCBPΔGIFΔ121 group, rGS14ΔCBPΔGIF group, GS14 group and PBS group on days 7 and 14 after challenging the wild-type strain on the inner thigh of the other side: (**I**) clinical signs of three random goats in the rGS14ΔCBPΔGIFΔ121 group were completely healed by day 14; (**II**) clinical signs of goats in the rGS14ΔCBPΔGIF group were only partially healed by day 14; (**III**) clinical signs of three random goats in the GS14 group showed scabs on day 7 and only partially healed by day 14; (**IV**) clinical signs of three random goats in the PBS group showed scabs and other signs on both day 7 and day 14. (**B**) Plots of changes in clinical scores after challenging in the rGS14ΔCBPΔGIFΔ121 group, rGS14ΔCBPΔGIF group, GS14 group and PBS group. Data for each group are expressed as mean standard error. (**C**) Plots of ORFV viral load in tissues at the challenged site and mouth and lip area on day 14 post challenging in the rGS14ΔCBPΔGIFΔ121 group, rGS14ΔCBPΔGIF group and PBS group. The data of each group are expressed as mean standard error. * *p* < 0.05, ** *p* < 0.01, and *** *p* < 0.001, **** *p* < 0.0001.

## Data Availability

The original contributions presented in the study are included in the article/Supplementary Material. Further inquiries can be directed to the corresponding author.

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
