# Peer review of "Construction of a Triple-Gene Deletion Mutant of Orf Virus and Evaluation of Its Safety, Immunogenicity and Protective Efficacy"

_vaccines, 2023, doi:10.3390/vaccines11050909_

Round 1

Reviewer 1 Report

“Construction of tripe-gene deletion mutant of Orf virus and evaluation on its safety, immunogenicity and immuno-protectivity”.

In this work, the authors generate an Orf virus lacking three genes that code for proteins that modulate the host's immune response. They analyze its safety and effectiveness as a vaccine and compare it with a recombinant double ORFV (previously characterized) and with the parent strain ORFV-GS14. The triple recombinant proved to be safe and to protect 100% of the animals against the challenge with a more virulent strain. These facts postulate it as a vaccine candidate.

Comments to the authors:

Table 2: it is already explained in the text, it is not necessary to add a table model, it can go as a supplementary figure.

Figure 3 F: the photographs must be improved, since it is not possible to appreciate details even by increasing the size of the image.

Figure 4: Please improve the contrast/brightness ratio of the images.

The Figure 6 A is the analysis strategy for CD4+ and CD8+ T-cells. I believe that these graphs should be smaller than Fig 6B and 6C where the results are shown.

Figure 8 C: The viral load is measured as Ct, in the area of the mouth and lips. But there are no photographs or score table of that area. What copy number are those Ct values equivalent to? Can it be correlated with a viral titer value?

Author Response

Response to Reviewer 1 Comments

Thank you for your valuable comments on my article, in response to which I have made the following changes:

Point 1: Table 2: it is already explained in the text, it is not necessary to add a table model, it can go as a supplementary figure. 

Response 1: I have removed the form from the text as you requested and uploaded it as additional material.

Point 2: Figure 3 F: the photographs must be improved, since it is not possible to appreciate details even by increasing the size of the image.

Response 2: In response to this revision, I combined the revision proposed by another reviewer and revised the entire Figure 3, where the original Figure 3 F was enlarged and renamed as Figure 2 B.

Point 3: Figure 4: Please improve the contrast/brightness ratio of the images.

Response 3: In response to this revision, I have redone Figure 4 once and increased the brightness of the entire image.

Point 4: The Figure 6 A is the analysis strategy for CD4+ and CD8+ T-cells. I believe that these graphs should be smaller than Fig 6B and 6C where the results are shown.

Response 4: I resized Figure 6A as you said to make sure it is smaller than the resulting image

Point 5: Figure 8 C: The viral load is measured as Ct, in the area of the mouth and lips. But there are no photographs or score table of that area. What copy number are those Ct values equivalent to? Can it be correlated with a viral titer value?

Response 5: In response to this revision, I reconverted the unit of measurement of CT values in the pictures to Lgcopies and redid the graphs. Among them, since the current inoculation site was mainly in the hairless area of the inner thigh, the study of the orofacial site was a small pre-experiment, so no pictures were taken, and the results of the orofacial site will be revealed as a future research direction in future experiments.

Reviewer 2 Report

1. The fact that it was not specified how many animals would be used in the trials under the title of material and method was considered as a deficiency.

2. While it is written that 18 animals will be used under the title of Safety test (128-132), it is seen that 16 animals are used within the groups. The reason for this difference is not understood.

3. Failure to give the doses of virus used (line 159) was considered a deficiency.

4. It would have been better to provide more detailed information on recombinant live attenuated vaccines in the introduction.

Author Response

Response to Reviewer 2 Comments

Thank you very much for your review of my article and your valuable comments, and I have made the following changes in response to your comments:

Point 1: 1. The fact that it was not specified how many animals would be used in the trials under the title of material and method was considered as a deficiency.

Response 1: In response to this revision, I added the total number of animals used to the Animals under Materials and Methods.

Point 2: 2. While it is written that 18 animals will be used under the title of Safety test (128-132), it is seen that 16 animals are used within the groups. The reason for this difference is not understood.

Response 2: I am very sorry I made a very cheap mistake, we bought 18 sheep in total, but two of them were used for other experiments and I forgot to exclude these two from the total number of sheep. I have now corrected the number in the text, thank you very much for your valuable input!

Point 3: Failure to give the doses of virus used (line 159) was considered a deficiency.

Response 3In response to this revision, I have added a new dose of virus to the text as you requested.

Point 4:  It would have been better to provide more detailed information on recombinant live attenuated vaccines in the introduction.

Response 4: Based on this revision, I have reinserted the description of recombinant live attenuated vaccines in the text.

Reviewer 3 Report

Line 17-20 suggest revision: “Previously, we have reported the construction of a double-gene deletion mutant of ORFV, (rGS14ΔCBPΔGIF), and its evaluation as a vaccine candidate. Building on this previous work, the current study reports the construction of a new vaccine candidate, generated by deleting a third gene (gene 121) to generate ORFV rGS14ΔCBPΔGIFΔ121. The in vitro growth characteristics, as well as the in vivo safety, immunogenicity, and protective efficacy, were evaluated.”

In line with this suggestion, I would also suggest the authors consider replacing “immuno-protectivity” with “protective efficacy” here and elsewhere in the manuscript, as appropriate.

Line 27 suggest revision “A virulent field strain of ORFV from an ORF scab was”

Line 41 suggest revision “and can result in secondary”

Line 27 suggest revision “ 138 kbp and encodes approximately 132 genes”

Line 49 suggest revision “and degree of conservation [13].”

line 50 suggest revision “The polypeptide encoded by the highly conservative gene, F1L, is frequently used in serological diagnosis of orf in sheep and goats [14].

Line 51 suggest revision: The abbreviation of “immune evasion regulatory proteins to IMPs, does not seem logical to me, I would suggest that IERPs is more appropriate.

Line 55 suggest revision: is to facilitate immune evasion by high affinity binding and inhibiting the production of cytokines

Line 62 suggest revision: “The conventional approach to controlling ORFV, is the prophylactic application of vaccination.

Line 67 While I would not question the risk of live vaccines and the potential for reversion to virulence. I cannot find any mention of reversion to virulence in the cited study. Please check for accuracy.

Line 69 suggest revision: “by deleting gene 121.”

Lines 89 to 92 – Please review this text. While I am very familiar with the approach used to construct the GFP transgene for the deletion of gene 121, I think that someone less familiar with the process would struggle to follow this text to replicate the described method.

Line 93 suggest revision: “of rGS14ΔCBPΔGIF

Line 105 The title of the table should be above the table. Consider providing this sample as a supplemental file. Also, consider adding the genomic coordinates for the ORFV specific primers with respect to GenBank accession.

Line 133 suggest replacing symptoms with signs

Line 138 This would appear to be more of a figure than a table. Could also be a supplemental file.

Line 159 suggest revision After 14 days, the goats

Line 162 Please check the citation Zz et al. in 2021

Line 162 suggest replacing homology with identity

Line 180 suggest replacing lesioned with cytopathic

Line 185 Figure 3 There are a lot of things within this figure.

Suggest that Panels 3A, 3B, 3E, 3D and 3G could be provided as supplemental files.

Line 213 suggest revision There was minimal inflammatory reaction

Line 215 suggest revision and then started exhibit scabs

Line 219 Figure legends should be below the panels.

Lines 221 to 223 suggest using alternative characters for the subpanels, eg Roman numerals instead of lowercase letters.

Line 226 What is a statistical line graph? Were these data tested for statistically significant differences?

Line 231 suggest revision clinical signs, and elsewhere in the manuscript where symptoms is used.

Line 246 Figure legends should be below the figure panels.

Line 267 suggest replacing causing with induces

What was the reason for not including rGS14ΔCBPΔGIF inoculated group in these analyses?

Line 280 suggest replacing immunization in with inoculation with

Immunization implies a beneficial effect, which not the case for the two control groups.

Line 283 suggest revision of the different treatment groups were taken 14 days post inoculation and the levels of IL-2 determined.

Similar to my previous point, immunization is not the appropriate term. In addition changes were not detected, as the measurements at each time point were analysed independently.

Line 284, 288 & 290 same as previously commented for Line 283

Line 292 The authors should review this heading and modify to something informative/meaningful.

In terms of the different viruses used in the study, I would suggest that authors consider more reader-friendly nomenclature for the different viruses used in the study.

For example;

rGS14ΔCBPΔGIFΔ121 could be rGS-TrypMut

rGS14ΔCBPΔGIF could be rGS12-DoubMut

GS14 could be GS12-wt

The acronyms could be defined when the viruses are described and should make the various treatment groups easier to discern throughout the manuscript.

Line 300 As mentioned before for the heading this type of shorthand is not appropriate, please describe your results.

Lines 300 to 305 it is generally not considered to be best practice to use cycle threshold (t) values directly in an analysis. It may be permissible if the amount of starting material used for template preparation can be standardised. For example, if it were a cell culture-based system a specific quantity of the culture supernatant. In this case where the starting material was a 10% suspensions of tissue samples (line 166), are the authors confident that the starting materials were consistent enough to enable direct comparison of Ct values? Did they consider using a reference gene for this purpose?

Line 389 suggest revision Th-1 like immune response and fully protect goats against the wild-type ORFV challenge. The prototype vaccine candidate elicited earlier, increased and sustained immune responses.

Author Response

Response to Reviewer 3 Comments

Thank you very much for reviewing my article in your busy schedule and making so many valuable comments. In response to your very valuable comments, I have made the following changes:

Point 1: Line 17-20 suggest revision: “Previously, we have reported the construction of a double-gene deletion mutant of ORFV, (rGS14ΔCBPΔGIF), and its evaluation as a vaccine candidate. Building on this previous work, the current study reports the construction of a new vaccine candidate, generated by deleting a third gene (gene 121) to generate ORFV rGS14ΔCBPΔGIFΔ121. The in vitro growth characteristics, as well as the in vivo safety, immunogenicity, and protective efficacy, were evaluated.”

In line with this suggestion, I would also suggest the authors consider replacing “immuno-protectivity” with “protective efficacy” here and elsewhere in the manuscript, as appropriate

Response 1: I have replaced "immuno-protectivity" with "protective efficacy" as you requested, and "immuno-protectivity" with "protective efficacy" elsewhere in the text, as appropriate.

Point 2: Line 27 suggest revision “A virulent field strain of ORFV from an ORF scab was”

Response 2: I have completed the revision according to the comments made.

Point 3: Line 41 suggest revision “and can result in secondary”

Response 3: I have completed the revision according to the comments made.

Point 4: Line 27 suggest revision “ 138 kbp and encodes approximately 132 genes”

Response 4: I have completed the revision according to the comments made.

Point 5: Line 49 suggest revision “and degree of conservation [13].”

Response 5: I have completed the revision according to the comments made.

Point 6: line 50 suggest revision “The polypeptide encoded by the highly conservative gene, F1L, is frequently used in serological diagnosis of orf in sheep and goats [14].

Response 6: I have completed the revision according to the comments made.

Point 7: Line 51 suggest revision: The abbreviation of “immune evasion regulatory proteins” to “IMPs”, does not seem logical to me, I would suggest that “IERPs” is more appropriate.

Response 7: I have completed the revision according to the comments made.

Point 8: Line 55 suggest revision: “is to facilitate immune evasion by high affinity binding and inhibiting the production of cytokines”

Response 8: I have completed the revision according to the comments made.

Point 9: Line 62 suggest revision: “The conventional approach to controlling ORFV, is the prophylactic application of vaccination.

Response 9: I have completed the revision according to the comments made.

Point 10: Line 67 While I would not question the risk of live vaccines and the potential for reversion to virulence. I cannot find any mention of reversion to virulence in the cited study. Please check for accuracy.

Response 10: In response to this revision, I have corrected the literature cited in the text.

Point 11:Line 69 suggest revision: “by deleting gene 121.”

Response 11: I have completed the revision according to the comments made.

Point 12: Lines 89 to 92 – Please review this text. While I am very familiar with the approach used to construct the GFP transgene for the deletion of gene 121, I think that someone less familiar with the process would struggle to follow this text to replicate the described method.

Response 12: In response to this revision, I have changed the textual descriptions in Lines 89 to 92 and re-designed the homologous recombination schematic to facilitate understanding.

Point 13: Line 93 suggest revision: “of rGS14ΔCBPΔGIF”

Response 13: I have completed the revision according to the comments made.

Point 14: Line 105 The title of the table should be above the table. Consider providing this sample as a supplemental file. Also, consider adding the genomic coordinates for the ORFV specific primers with respect to GenBank accession.

Response 14: In response to this revision, I have removed the table and submitted it as supplementary material, and GenBank has marked it in the text.

Point 15: Line 133 suggest replacing “symptoms” with “signs”

Response 15: I have completed the revision according to the comments made.

Point 16:Line 138 This would appear to be more of a figure than a table. Could also be a supplemental file.

Response 16: I have removed the form and submitted it as a supplement.

Point 17:Line 159 suggest revision “After 14 days, the goats”

Response 17: I have completed the revision according to the comments made.

Point 18: Line 162 Please check the citation “Zz et al. in 2021”

Response 18: After checking, I changed the citation "Zz et al. in 2021" to "Zz et al. in 2022".

Point 19: Line 162 suggest replacing “homology” with “identity”

Response 19: I have completed the revision according to the comments made.

Point 20:Line 180 suggest replacing “lesioned” with “cytopathic”

Response 20: I have completed the revision according to the comments made.

Point 21: Line 185 Figure 3 There are a lot of things within this figure.

Suggest that Panels 3A, 3B, 3E, 3D and 3G could be provided as supplemental files.

Response 21: I have revised it according to your suggestion: the original Figure 3A,3B,3E,3D and 3G are deleted and uploaded as supplementary materials, and the original Figure 3C and 3F are renamed as Figure 2A and 2B and kept in the text.

Point 22: Line 213 suggest revision “There was minimal inflammatory reaction”

Response 22: I have completed the revision according to the comments made.

Point 23: Line 215 suggest revision “and then started exhibit scabs”

Response 23: I have completed the revision according to the comments made.

Point 24: Line 219 Figure legends should be below the panels.

Response 24: I have completed the changes as you requested, and have also made changes to the other legends.

Point 25: Lines 221 to 223 suggest using alternative characters for the subpanels, eg Roman numerals instead of lowercase letters.

Response 25: I have completed the revision according to the comments made.

Point 26: Line 226 What is a “statistical line graph”? Were these data tested for statistically significant differences?

Response 26: Regarding this comment, I would like to explain and clarify it: because I wanted to show the trendiness of the overall clinical score changes, I connected the clinical scores at each time point with a dash line. These data were statistically analyzed for variability, but since I did not know where to label the variability, I did not show it in the image. Based on your revision, I redid the bar chart with variability analysis and replaced the original line statistical chart in the text.

Point 27: Line 231 suggest revision “clinical signs”, and elsewhere in the manuscript where “symptoms” is used.

Response 27: I have completed the corrections you requested and made other changes to the text where "symptoms" is concerned.

Point 28: Line 246 Figure legends should be below the figure panels.

Response 28I have completed the revision according to the comments made.

Point 29:Line 267 suggest replacing “causing” with “induces”

What was the reason for not including rGS14ΔCBPΔGIF inoculated group in these analyses?

Response 29I have completed revisions to the comments made. Regarding the reason for not including rGS14ΔCBPΔGIF inoculated group in these analyses, I need to explain as follows: The safety experiments are the result of aggregation of several animal experiments, and since rGS14ΔCBPΔGIF was involved in the previous studies and was not well protected in the pre-experiments, we did not design the rGS14ΔCBPΔGIF group in the animal experiments involving cytokine detection for animal welfare and animal ethics reasons. For animal welfare and animal ethical reasons, we did not design the rGS14ΔCBPΔGIF group in animal experiments involving cytokine detection, and therefore did not analyze the rGS14ΔCBPΔGIF group.

Point 30:Line 280 suggest replacing “immunization in” with “inoculation with”

Immunization implies a beneficial effect, which not the case for the two control groups.

Response 30: I have completed the revision according to the comments made.

Point 31: Line 283 suggest revision “of the different treatment groups were taken 14 days post inoculation and the levels of IL-2 determined.”

Similar to my previous point, immunization is not the appropriate term. In addition “changes” were not detected, as the measurements at each time point were analysed independently.

Response 31: I have completed the revision according to the comments made.

Point 32: Line 284, 288 & 290 – same as previously commented for Line 283

Response 32: I have completed the revision according to the comments made.

Point 33: Line 292 The authors should review this heading and modify to something informative/meaningful.

In terms of the different viruses used in the study, I would suggest that authors consider more reader-friendly nomenclature for the different viruses used in the study.

For example;

rGS14ΔCBPΔGIFΔ121 could be rGS-TrypMut

rGS14ΔCBPΔGIF could be rGS12-DoubMut

GS14 could be GS12-wt

The acronyms could be defined when the viruses are described and should make the various treatment groups easier to discern throughout the manuscript.

Response 33: I have completed the revision according to the comments made.

Point 34: Line 300 As mentioned before for the heading this type of “shorthand” is not appropriate, please describe your results.

Response 34: I have completed the revision according to the comments made.

Point 35: Lines 300 to 305 – it is generally not considered to be best practice to use cycle threshold (t) values directly in an analysis. It may be permissible if the amount of starting material used for template preparation can be standardised. For example, if it were a cell culture-based system a specific quantity of the culture supernatant. In this case where the starting material was a “10% suspensions of tissue samples” (line 166), are the authors confident that the starting materials were consistent enough to enable direct comparison of Ct values? Did they consider using a reference gene for this purpose?

Response 35: Based on this observation, I replaced the originally used Ct values with Lgcopies.In this experiment, I ground the skin tissues at a ratio of weight:tissue grinding solution = 1:10 to minimize the effect of errors.

Point 36:Line 389 suggest revision “Th-1 like immune response and fully protect goats against the wild-type ORFV challenge. The prototype vaccine candidate elicited earlier, increased and sustained immune responses.”

Response 36: I have completed the revision according to the comments made.

Round 2

Reviewer 3 Report

The authors have adequately addressed the comments and suggestions I made during my initial review of their manuscript.

I have one additional minor comment.

I asked the authors to check the citation “Zz et al. in 2021" and it was revised to “Zz et al. in 2022".

However, there is no reference by “Zz et al." in the provided reference list. I believe the reference should be "Zhu et al. [3]." 

Author Response

Response to Reviewer 3 Comments

Thank you very much for reviewing my article in your busy schedule and making so many valuable comments. In response to your very valuable comments, I have made the following change:

Point 1:

I asked the authors to check the citation “Zz et al. in 2021" and it was revised to “Zz et al. in 2022".

However, there is no reference by “Zz et al." in the provided reference list. I believe the reference should be "Zhu et al. [3]." 

Response 1:Thank you very much for your valuable comments, I have re-made the changes as you requested.
